# Blockade of LIF and PD-L1 Enhances Chemotherapy in Preclinical PDAC Models

**DOI:** 10.3390/cancers17020204

**Published:** 2025-01-09

**Authors:** Jian Ye, Shuyang S. Qin, Angela L. Hughson, Gary Hannon, Tara G. Vrooman, Maggie L. Lesch, Sarah L. Eckl, Lauren Benoodt, Bradley N. Mills, Edith M. Lord, Brian A. Belt, David C. Linehan, Nadia Luheshi, Jim Eyles, Scott A. Gerber

**Affiliations:** 1Department of Surgery, University of Rochester Medical Center, Rochester, NY 14642, USA; jian_ye@urmc.rochester.edu (J.Y.); shuyang_qin@urmc.rochester.edu (S.S.Q.); angie_hughson@urmc.rochester.edu (A.L.H.); gary_hannon@urmc.rochester.edu (G.H.); tara_vrooman@urmc.rochester.edu (T.G.V.); maggie_lesch@urmc.rochester.edu (M.L.L.); sarah_eckl@urmc.rochester.edu (S.L.E.); bradley_mills@urmc.rochester.edu (B.N.M.); brian_belt@urmc.rochester.edu (B.A.B.); david_linehan@urmc.rochester.edu (D.C.L.); 2Center for Tumor Immunology Research, University of Rochester Medical Center, Rochester, NY 14642, USA; edith_lord@urmc.rochester.edu; 3Wilmot Cancer Institute, University of Rochester Medical Center, Rochester, NY 14627, USA; 4Department of Microbiology and Immunology, University of Rochester Medical Center, Rochester, NY 14642, USA; 5Genomic Research Center, University of Rochester Medical Center, Rochester, NY 14642, USA; lauren_benoodt@urmc.rochester.edu; 6Oncology R&D, AstraZeneca, Aaron Klug Building, Granta Park, Cambridge CB2 0AA, UK; nadia.luheshi@astrazeneca.com (N.L.); jim.eyles@astrazeneca.com (J.E.); 7Department of Radiation Oncology, University of Rochester Medical Center, Rochester, NY 14626, USA

**Keywords:** pancreatic ductal adenocarcinoma (PDAC), chemotherapy, leukemia inhibitory factor (LIF), programmed death-ligand 1 (PD-L1)

## Abstract

LIF has emerged as a key target in pancreatic cancer (PDAC) due to its role in promoting chemoresistance by driving tumor progression and promoting an immunosuppressive tumor microenvironment. Early clinical trial data testing anti-LIF treatment in patients proved safe but with limited efficacy, suggesting the need to be combined with other therapies such as chemotherapy or immunotherapy. The aim of this study was to assess the antitumor efficacy and mechanism of action resulting from combination therapy consisting of chemotherapy (gemcitabine/nab-paclitaxel), anti-LIF, and anti-PD-L1 in mouse models of PDAC. We demonstrated that this triple combination therapy significantly improved antitumor efficacy compared to monotherapy or dual combinations of these therapies. Mechanistically, this combination therapy not only reduced the aggressiveness of tumor cells but also enhanced antitumor immunity, especially through CD8 T cells. Our preclinical data provide the rationale for ongoing or future clinical trials investigating the combination of anti-PD-L1 with anti-LIF and chemotherapy in PDAC.

## 1. Introduction

Pancreatic cancer is a highly aggressive and deadly disease, with PDAC being the common histopathologic type, representing over 90% of cases. In 2024, it is projected that 66,440 new cases of pancreatic cancer and 51,570 related deaths will occur in the USA [1]. Currently, pancreatic cancer ranks as the third leading cause of cancer-related deaths, after lung and colon cancers, and it is expected to become the second leading cause before 2030 [2]. Despite decades of extensive research and clinical trials, the prognosis for PDAC remains poor, with the 5-year survival rate only improving slightly to 13%. Surgical resection remains the sole curative option for PDAC, but due to the lack of early symptoms and effective screening, up to 80% of patients are initially diagnosed with advanced or metastatic disease, making them ineligible for surgery. Chemotherapy regimens, such as FOLFIRINOX or gemcitabine/nab-paclitaxel, are essential for treating the majority of PDAC patients [3,4]. However, PDAC frequently develops resistance to these therapies, highlighting the urgent need for more effective and durable therapies. Whereas immunotherapy has shown success in other cancers [5,6,7], multiple trials have shown that PDAC is less responsive to such treatments [8,9], emphasizing the need for additional interventions to overcome the immunosuppressive environment in PDAC.

Leukemia inhibitory factor (LIF), a member of the interleukin-6 (IL-6) cytokine family, was first recognized for inducing differentiation in myeloid leukemia cells [10,11]. It is now known to play a key role in cancer progression by regulating cell differentiation, renewal, and survival [12,13,14]. Clinically, elevated LIF expression correlates with poor prognosis and reduced responsiveness to anti-cancer treatment [12,15,16]. Recent clinical and preclinical studies in PDAC point to serum LIF as a promising biomarker for diagnosis, disease monitoring, and prognosis, offering greater sensitivity and specificity than traditional markers like CA19-9 and CEA [13].

Research on the role of LIF in PDAC has primarily focused on aspects of tumor biology, including tumor stemness, EMT, growth, and survival [13,14,17,18]. However, LIF also has significant immunoregulatory functions. Physiologically, LIF promotes immunosuppression by facilitating the differentiation of immunosuppressive regulatory T cells (Treg), semi-mature dendritic cells (DCs), and M2 macrophages, while inhibiting Th17 cells, mature DCs, and M1 macrophages [11]. LIF has been identified as a novel predictive biomarker for resistance to PD1/PD-L1 blockade in non-small cell lung cancer (NSCLC) and bladder cancer patients, highlighting the potential of targeting LIF to enhance immune checkpoint blockade (ICB) therapy [19]. A recent study has shown that LIF regulates CD8^+^ T cell tumor infiltration by repressing CXCL9 expression in tumor-associated macrophages, and blocking LIF enhances T cell infiltration, improving responses to anti-PD1 therapy in glioblastoma and ovarian cancer models [20]. Despite these findings, the immunoregulatory role of LIF in PDAC and the potential benefits of LIF blockade in improving immunotherapy remain unclear.

Inhibiting LIF presents a promising therapeutic approach due to its potential to affect various cancer mechanisms. Neutralizing antibodies and small-molecule inhibitors targeting the LIF/LIFR pathway have been developed for preclinical research and clinical trials [15,17,21,22,23,24,25]. Among these, MSC-1 (also known as AZD0171), a humanized monoclonal antibody, has shown strong and specific LIF antagonism in multiple cancer types [25]. Although a phase 1 clinical trial of MSC-1 demonstrated tolerability and safety as a monotherapy, the limited efficacy of LIF [26] suggests that targeting this molecule alongside standard treatments, such as chemotherapies and/or immunotherapies, could be a promising approach.

In this study, using an orthotopic PDAC murine model, we demonstrated that sequential treatment with chemotherapy (gemcitabine/nab-paclitaxel) and dual blockade of LIF and PD-L1 significantly enhanced antitumor efficacy compared to monotherapy or the dual combination of these therapies. The chemo/anti-LIF/anti-PD-L1 treatment decreased mesenchymal tumor cells and augmented the antitumor immune response, primarily driven by CD8 T cells, as depleting these cells diminished the treatment’s effectiveness. It also repolarized macrophages towards antitumor phenotypes. Our preclinical data, combined with findings from other studies, strongly advocate for advancing clinical trials that target LIF in combination with standard-of-care therapies for PDAC.

## 2. Materials and Methods

### 2.1. Cells and Reagents

The murine PDAC KCKO and luciferase-expressing KCKO (KCKO-Luc) cell line was generously provided by Dr. Pinku Mukherjee from the University of North Carolina, Charlotte, NC. The cell line was confirmed to be free of mycoplasma and maintained in RPMI1640 supplemented with 10% FBS and 1% penicillin-streptomycin.

Anti-LIF (mAZD0171), anti-PD-L1 (Clone 80), and corresponding isotype IgG controls are provided by AstraZeneca. For in vivo experiments, anti-LIF (20 mg/kg), anti-PD-L1 (10 mg/kg), or the respective isotype control IgG were administered via intraperitoneal injections (diluted in PBS, 100 μL per mouse) twice per week starting 3 days after chemotherapy or concurrently with chemotherapy. To deplete CD4^+^ or CD8^+^ T cells, 200 μg of anti-CD4 (clone: YTS191), anti-CD8 (clone: 53–6.7), or isotype IgG (diluted in PBS, 100 µL per mouse) were administered intraperitoneally every 3 days. For inflammatory monocyte depletion, mice were given CCR2i (RS504393, TOCRIS) 2 mg/kg, intraperitoneally.

### 2.2. Murine Orthotopic Model, KPC Model, and Chemotherapy

Female C57BL/6J mice (6 to 8 weeks old) were purchased from Jackson Laboratory and acclimated in the institutional animal facility. All animal studies were approved by the University Committee on Animal Resources (UCAR) at the University of Rochester Medical Center (Rochester, NY, USA). Orthotopic models were established as previously described, with minor modifications [27,28]. Briefly, mice were anesthetized with isoflurane, and 2 × 10^5^ KCKO-Luc cells suspended in a 1:1 mixture of PBS to Matrigel were injected into the tail of the pancreas. Mice were treated with gemcitabine (100 mg/kg, intraperitoneally) and nab-paclitaxel (100 mg/kg, via tail vein injection) on days 5 and 9 following tumor implantation. *P48-Cre*^+/−^; *Tp53^L/L^* and *LSL-Kras^G12D^* mice, obtained from Dr. Aram Hezel, were crossed to generate the *P48-Cre*; *LSL-Kras^G12D^*; *Tp53^L/L^* (KPC) genotype [29]. Six- to seven-week-old KPC mice were treated with two doses of chemotherapy (gemcitabine + nab-paclitaxel) as KCKO-Luc orthotopic models.

### 2.3. Flow Cytometry

Mouse tumors were minced into small fragments and digested with 30% collagenase for 30 min at 37 °C. Single-cell suspensions were prepared by passing tumor fragments through a 40 mm cell strainer after suspension in PBS containing 5% FBS. To prevent non-specific binding, cells were incubated with Fc receptor blocking solution. For cell surface staining, fluorophore-conjugated mouse antibodies (BD Biosciences or BioLegend) were added and incubated for 30 min at 4 °C in the dark. The antibodies used included anti-CD45-FITC(clone 30-F11), anti-CD45-PE-Cy5(clone 30-F11), anti-CD3-FITC(clone 145-2C11), anti-CD4-APC-Cy5(clone GK1.5), anti-CD8-BV605(clone 53-6.7), anti-CD11b-eFluor450(clone M1/70), anti-F4/80-APC(clone BM8), anti-Ly6C-APC-Cy7(clone AL-21), anti-Ly6G-PE-Cy7(clone 1A8), anti-CD11c-PE-Cy7(clone HL3), anti-MHCII(I-A/I-E)-PerCP-Cy5.5(clone M5/114.15.2), anti-CD69-APC(clone H1.2F3), anti-PD-L1-BV605(clone 10F.9G2), anti-CD24-BV605(clone M1/69), anti-CD103-APC(clone M290), anti-XCR1-BV421(clone ZET), anti-CD80-BV786(clone 16-10A1), anti-CD86-BV510(clone PO3), anti-PD1-BV711(clone J43), or anti-CTLA4-APC-R700(clone UC10-4F10-11).

For further intracellular staining, cells were stimulated with PMA and ionomycin in the presence of GolgiStop (ThermoFisher, Waltham, MA, USA, cat# 00-4975-93) for 5 h. Cells were then washed with PBS containing 5% FBS, fixed, and permeabilized with a FoxP3/Transcription factor staining buffer set (eBiosciences, Santa Clara, CA, USA) and stained with fluorescence-labeled mouse antibodies, including anti-IFNg-BV786(clone XMG1.2), anti-GzmB-Pacific Blue (clone GB11), and anti-FoxP3-APC (clone FJK-16s), for 30 min at room temperature in the dark. Stained cells were washed and resuspended in PBS containing 5% FBS. Flow cytometry analysis was conducted using an LSRII, and data were analyzed with FlowJo software (v10.10.0).

### 2.4. ScRNA-Seq and Analysis

Tumor tissues from mice were dissociated using the Tumor Dissociation Kit (Miltenyi Biotec 130-096-730) following the protocol from 10× Genomics (CG00147.Rev B). Suspended single cells were stained with Ghost dye violet 510 and sorted for live cells (unstained). Cells from different samples were multiplexed, captured, and sequenced using 10× Genomics protocols at the University of Rochester Medical Center (URMC) genomic research center (GRC).

Raw scRNA-seq data were processed with 10× Genomics CellRanger software (v6.0.2) to perform demultiplexing and derive raw count values for each cell captured using the mm10 reference. Seurat (version 5.1.0) installed in RStudio (version 4.3.2) was employed for quality control, cell filtering, data normalization, cell clustering, and identification of cluster marker genes. For quality control, cells with more than 10% mitochondrial genome content were excluded, and a minimum of 200 and a maximum of 7000 genes per cell were set as thresholds to remove low-quality cells or doublets. Data normalization was conducted using the “SCTransform” function. Dimensionality reduction and cell clustering were performed via the “RunPCA” and “RunUMAP” functions. The signature genes for each cell cluster were identified using the “FindAllmarkers” function. The differentially expressed genes (DEGs) for selected clusters between different treatment groups were identified by the “FindMarker” function. “VlnPlot”, “DotPlot”, “FeaturePlot”, and “RidgePlot” functions were employed to display the expression profiles of genes of interest in different cell clusters or between different treatment groups. Genes with adjusted *p*-value < 0.05 and log2FoldChange > 0.25 or <−0.25 were subjected to Enrichr analysis to identify significantly enriched upregulated or downregulated pathways. The average expression of upregulated or downregulated genes from relevant signal pathways was further calculated using the “AddModuleScore” function and depicted in RidgePlot or VlnPlot across the different treatment groups.

### 2.5. Statistical Analysis

Statistical analyses were conducted using GraphPad Prism 9.3.1 software. Unless otherwise specified, data are expressed as means ± SEM. For multiple comparisons of growth curves, one-way ANOVA was performed, followed by a Bonferroni adjustment. Survival curve multiple comparison was analyzed using the log-rank (Mantel–Cox) test with Bonferroni adjustment. For multiple group comparisons, one-way ANOVA was utilized, followed by the Dunnett test to compare experimental groups against the untreated group. For single comparisons between the two groups, unpaired student *t*-tests were employed. Statistical significance was defined as *p* < 0.05 or *p* < 0.01.

## 3. Results

### 3.1. Sequential Treatments with Chemotherapy and LIF/PD-L1 Blockade Reduce PDAC Tumor Burden and Promote Survival

In the KCKO-Luc orthotopic mice model of PDAC, which harbors KRAS mutation [27,30,31], we evaluated the efficacy of combination treatment with chemotherapy, anti-LIF, and anti-PD-L1. As shown in Figure 1a, tumor-bearing mice were treated with two doses of standard of care gemcitabine/nab-paclitaxel, followed by twice-weekly doses of anti-LIF and anti-PD-L1. Monotherapy with anti-LIF, anti-PD-L1, or chemo did not significantly control tumor growth. However, both chemo+anti-PD-L1 and chemo+anti-LIF+anti-PD-L1 treatments significantly reduced tumor growth compared to untreated controls. Notably, tumor control was markedly improved in the chemo/anti-LIF/anti-PD-L1 group compared to the chemo/anti-PD-L1 group (Figure 1b). Both chemo/anti-PD-L1 and chemo/anti-LIF/anti-PD-L1 treatments also led to improved survival relative to the untreated group, but only the chemo/anti-LIF/anti-PD-L1 group showed a survival benefit compared to chemotherapy alone, whereas the chemo/anti-PD-L1 group did not (Figure 1c). This was confirmed in the KPC mice model, a spontaneous PDAC model, where anti-LIF/anti-PD-L1 treatment significantly improved survival compared to chemotherapy alone (Appendix A, Figure A1). No significant toxicity was observed, as demonstrated by the lack of significant body weight changes in any treatment group compared to the untreated animals (Appendix A, Figure A2). Additionally, we explored whether starting anti-LIF treatment earlier (on day 5 post tumor implantation, along with chemotherapy, Appendix A, Figure A3a) would enhance efficacy. Surprisingly, this concurrent treatment schedule did not improve the antitumor efficacy of chemo/anti-PD-L1 based on tumor growth and survival data (Appendix A, Figure A3b,c). Together, sequential rather than concurrent treatment with chemotherapy and LIF/PD-L1 blockade significantly enhanced antitumor efficacy.

### 3.2. Chemo/Anti-LIF/Anti-PD-L1 Enhances the Antitumor Immune Response in an Orthotopic Model of Murine PDAC

#### 3.2.1. Chemo/Anti-LIF/Anti-PD-L1 Induces Increased Infiltration of Functional CD8^+^ T Cells

To evaluate the impact of chemo/anti-LIF/anti-PD-L1 on the tumor immune microenvironment, we followed the same experimental design as in Figure 1a and assessed the tumor-infiltrating immune cells using flow cytometry at day 15 post tumor implantation. We selected Day 15 to capture early immune responses preceding tumor volume changes and to ensure sufficient tissue availability for flow cytometry analysis. The chemo/anti-LIF/anti-PD-L1 treatment resulted in an enhanced infiltration of CD8^+^ T cells, GzmB^+^CD8^+^ T cells, and PD1^+^CD8^+^ T cells (Figure 2a). Additionally, we observed a trend of increased infiltration of CD4^+^ T cells, CD4^+^CD69^+^ T cells, and PD1^+^CD4^+^ T cells (Figure 2b). Pathway enrichment analysis from scRNA-seq data (Appendix A, Figure A4 and Figure A5) further supported these findings, showing upregulation of effector function genes, including interferon gamma/alpha response signaling, in both CD8 and CD4 T cells, as well as increased expression of proliferation-related pathways, such as E2F targets and G2-M in CD8 T cells following chemo/anti-LIF/anti-PD-L1 treatment.

#### 3.2.2. Chemo/Anti-LIF/Anti-PD-L1 Modulates Myeloid Cells Toward a Proinflammatory and Anti-Tumor Phenotype

We investigated populations of myeloid cells and observed an increase in monocytes and a decrease in granulocytes within the tumor following chemo/anti-LIF/anti-PD-L1 (Figure 2c). We further examined the expression of activation/maturation markers in myeloid cells and observed a significant increase in MHCII on monocytes, as well as a trend of increased CD86 on TAMs, following chemo/anti-LIF/anti-PD-L1 treatment (Appendix A, Figure A6). Since LIF is known to influence macrophage differentiation [32,33], we further investigated changes in macrophages post-treatment. Monocyte/macrophage cells (Appendix A, Figure A4, clusters 0, 1, and 4 from the original UMAP) were selected for re-clustering analysis. These clusters revealed distinct populations of monocytes, M1 macrophages, and M2 macrophages, identified by canonical marker genes (Appendix A, Figure A7). Treatment with chemo/anti-LIF/anti-PD-L1 resulted in an increase in monocytes and M1 macrophages, while M2 macrophages decreased. Subsequent pathway enrichment analysis demonstrated upregulation of interferon response pathways and downregulation of hypoxia pathways and metabolism-related pathways, including cholesterol homeostasis and mTORC1 signaling. Together, these findings suggest that chemo/anti-LIF/anti-PD-L1 treatment modulates monocytes/macrophages, promoting a shift toward a proinflammatory, anti-tumor phenotype.

DC clusters (Flt3^+^, cluster 5, 13, and 16 from the original UMAP, Appendix A, Figure A4a) were also re-clustered and identified as MoDC, cDC1, cDC2, mregDC, and pDC based on their canonical marker genes. Treatments with chemo, chemo/anti-PD-L1, or chemo/anti-LIF/anti-PD-L1 led to an increase in both cDC2 and pDC populations (Appendix A, Figure A8a–d). Similarly to the effect on macrophages, chemo/anti-LIF/anti-PD-L1 treatment upregulated interferon response pathways (Appendix A, Figure A8e,f), indicating DC maturation and activation, while downregulating metabolism-related pathways, including mTORC1, cholesterol homeostasis, and glycolysis (Appendix A, Figure A8g,h).

#### 3.2.3. CD8 Depletion Abolished the Antitumor Efficacy of Chemo/Anti-LIF/Anti-PD-L1

To further investigate the role of CD8^+^ T cells, CD4^+^ T cells, and monocytes, we depleted these cells using antibodies against CD8, CD4, or CCR2 inhibitors, respectively (Figure 3a). We determined that CD8 depletion largely, though not completely, abolished the antitumor efficacy of chemo/anti-LIF/anti-PD-L1. CD4 depletion only partially reduced the efficacy, whereas monocyte depletion had no effect (Figure 3b). Similarly, CD8 depletion, but not CD4 or monocyte depletion, reversed the survival benefit of chemo/anti-LIF/anti-PD-L1 treatment (Figure 3c). Therefore, chemo/anti-LIF/anti-PD-L1 enhanced the antitumor immune response, with CD8^+^ T cells playing a key role.

### 3.3. Single-Cell RNA-Sequencing (scRNA-Seq) Analysis Reveals Decrease in Mesenchymal Tumor Cells After Chemo/Anti-LIF/Anti-PD-L1 Treatment

Given that LIF has been implicated in EMT in pancreatic cancer [13], we sought to determine whether chemo/anti-LIF/anti-PD-L1 affects EMT in our model. Clusters 20 and 12 from the original UMAP (Appendix A, Figure A4b) were identified as PDAC tumor cells based on their expression of marker genes, including Sox9, Krt7, Krt19, or Epcam (Figure 4a) [34]. Copy number variation (CNV) analysis using inferCNV, which identifies genomic alterations involving gains or losses of DNA segments, further validated the tumor cell identification, revealing significant genomic alterations in clusters 12 and 20 (Appendix A, Figure A9). Specifically, cluster 12 exhibited higher CNVs, consistent with its characterization as mesenchymal-like tumor cells, whereas cluster 20 displayed fewer CNVs, consistent with its epithelial-like phenotype. In support of this, EMT scores were calculated as the mean expression of the mesenchymal markers subtracted by the mean expression of the epithelial markers as previously described [35,36]. As shown in the violin plots (Figure 4b), chemo/anti-LIF/anti-PD-L1 significantly decreases EMT in tumor cells.

Subcluster analysis revealed a distinct cluster 1, primarily corresponding to original cluster 20 in Appendix A, Figure A4b, which exhibited high expression of epithelial markers such as Cdh1, F11r, Ctnnd1, and Epcam. In contrast, clusters 0 and 2–7, mainly corresponding to original cluster 12 in Figure A3a, showed elevated expression of mesenchymal markers including Vim, Zeb1, Fn1, and Col6a1 (Figure 4c,d). Treatments significantly reduced the mesenchymal cell population, with the chemo/anti-LIF/anti-PD-L1 group showing the most reduced levels (Figure 4e,f). Further signal pathway analysis in tumor cells revealed that the EMT pathway is the most significantly downregulated pathway following chemo/anti-LIF/anti-PD-L1 treatment (Figure 4g,h). We also observed other significant downregulated pathways, including the stem cell-related pathway Myc Targets V1, hypoxia, and glycolysis (Figure 4g). Using genes associated with previously reported classifications of classical and basal-like tumor cells [37,38,39], we determined that chemo/anti-LIF/anti-PD-L1 significantly decreased genes associated with basal-like tumors (Appendix A, Figure A10), consistent with the observed downregulation of EMT.

## 4. Discussion

LIF has emerged as a key target in PDAC due to its role in promoting chemoresistance by enhancing stemness and EMT [13]. However, its potential role in modulating immunosuppression in PDAC remains unclear. In this study, we demonstrated that combining chemotherapy with anti-LIF and anti-PD-L1 treatments significantly enhances antitumor efficacy compared to chemotherapy alone or in combination with anti-PD-L1. Mechanistically, this combination therapy not only reduces the number of mesenchymal tumor cells but also induces an antitumor immune response, marked by an increase in functional CD8^+^ T cells, which are crucial for antitumor activity. Additionally, the combination therapy shifts macrophages and dendritic cells towards antitumor phenotypes.

LIF has been shown to promote tumor cell stemness, EMT, growth, and survival, contributing to chemo/radiotherapy resistance in various tumor types, including PDAC (13,14,16,17). Consistent with these studies, our scRNA-seq analysis reveals a significant decrease in tumor EMT and stemness (Myc Targets V1 signaling) following chemo/anti-LIF/anti-PD-L1 treatment (Figure 4b,e–h). Our analysis also reveals a downregulation of glycolysis in tumor cells following treatment (Figure 4g). LIF has been reported to drive metabolic reprogramming in cancer cells, particularly by promoting glycolysis and thereby fostering tumorigenesis in breast cancer. Mechanistically, LIF activates the AKT pathway, which facilitates the translocation of the glucose transporter Glut1 to the plasma membrane, enhancing glucose uptake and promoting glycolysis in cancer cells [40]. Whether a similar mechanism, where LIF regulates glycolysis in tumor cells, exists in PDAC remains to be elucidated, as we observed a downregulation of glycolysis following treatment. Additionally, we observed downregulation of hypoxia signal in tumor cells following treatments (Figure 4g). LIF has been shown to regulate vascularization, with Lif(−/−) mice exhibiting increased microvessel density in various tissues [41]. Whether anti-LIF exerts a similar effect by modulating vascularization to reduce hypoxia in PDAC remains to be investigated.

PDAC is marked by a highly immunosuppressive TME, largely driven by an abundance of suppressive myeloid cells. LIF has been increasingly linked to enhanced ICB when inhibited [19,20]. However, the specific immunoregulatory role of LIF in PDAC remains largely unstudied. In our study, we demonstrate that chemo/anti-LIF/anti-PD-L1 significantly reshapes the immune TME, with several key findings: First, we observed a marked increase in CD8^+^ T cell infiltration, particularly Gzmb^+^ CD8^+^ T cells, accompanied by upregulated interferon alpha/gamma response and cell cycle-related pathways in scRNA-seq data. CD8 depletion experiments further showed that the antitumor efficacy of the treatment is primarily CD8^+^ T cell dependent. Second, anti-LIF repolarized monocytes and macrophages toward an anti-tumor phenotype. Whereas previous studies have shown that targeting suppressive monocytes in PDAC with a CCR2 inhibitor could enhance chemotherapy [31,42], in our current study, inhibition of monocytes from entering the tumor with a CCR2 inhibitor did not significantly affect the chemo/anti-LIF/anti-PD-L1 response. This suggests that the treatment may partially reverse the immunosuppressive function of these cells. Third, chemo/anti-LIF/anti-PD-L1 also promotes DC maturation/activation, suggesting it enhances tumor antigen presentation and ultimately triggers the activation of effector CD8 T cells. Together, these findings underscore the potential of combining anti-LIF with chemotherapy and immunotherapy to transform the immunosuppressive TME in PDAC, creating a more immunostimulatory environment that enhances antitumor immune responses.

The combination treatment of chemotherapy, anti-LIF, and anti-PD-L1 led to an increase in M1 macrophages and a reduction in M2 macrophages, accompanied by enhanced interferon response pathways, suggesting a shift towards an anti-tumor phenotype. Supporting our findings, LIF has been linked to the gene signature of suppressive TAMs in various tumors, including PDAC [20,25]. Neutralizing antibodies against LIF have driven TAMs to acquire antitumor and proinflammatory functions, and combining anti-LIF with anti-PD1 results in a strong antitumor response in colon cancer, GBM, and ovarian cancer [20,25]. Additionally, previous studies have shown that macrophages undergo a metabolic shift induced by PDAC cells, transitioning from oxidative phosphorylation to glycolysis [43,44,45]. This shift promotes the polarization of macrophages towards the M2 phenotype, which supports tumor growth and invasion. Therefore, the observed decrease in M2 macrophages following triple therapy may be due to the downregulation of metabolism-related signals, particularly glycolysis, in macrophages.

In summary, our findings suggest that the combination of chemotherapy, anti-LIF, and anti-PD-L1 synergistically enhances antitumor activity in PDAC through both tumor-intrinsic and immune-mediated mechanisms. Tumor-intrinsic effects include the reduction in mesenchymal tumor cells, stemness, and glycolysis, accompanied by a downregulation of hypoxia signals. Immune-mediated mechanisms involve the reshaping of the immunosuppressive TME, including increased infiltration and activation of functional CD8^+^ T cells, repolarization of macrophages and monocytes toward antitumor phenotypes, and enhanced dendritic cell maturation and antigen presentation. Interestingly, administering anti-LIF earlier, alongside chemotherapy, may disrupt the recruitment and activation of key immune cells, such as macrophages and effector T cells, which are essential for mounting an effective antitumor response. This highlights the importance of optimal treatment sequencing to fully leverage the immune-stimulating potential of anti-LIF. Together, these effects transform the TME into a more immunostimulatory environment, reversing immunosuppression and enhancing CD8^+^ T cell-mediated antitumor responses.

## 5. Conclusions

In summary, the combination of chemotherapy, anti-LIF, and anti-PD-L1 not only targets tumor cells but also modulates the broader tumor immune response. Our data suggest a potential signature for anti-LIF treatment, characterized by reduced tumor stemness, proliferation, and metabolism-related signaling. Additionally, there is increased infiltration of effector CD8 cells, M1 macrophages, pDC, and cDC2, along with enhanced activation/maturation signaling in tumor-infiltrating CD8 cells, macrophages, and dendritic cells. These findings provide the rationale for ongoing or future clinical trials investigating the combination of ICB with anti-LIF and chemotherapy in PDAC (NCT04999969).

## Figures and Tables

**Figure 1 cancers-17-00204-f001:**
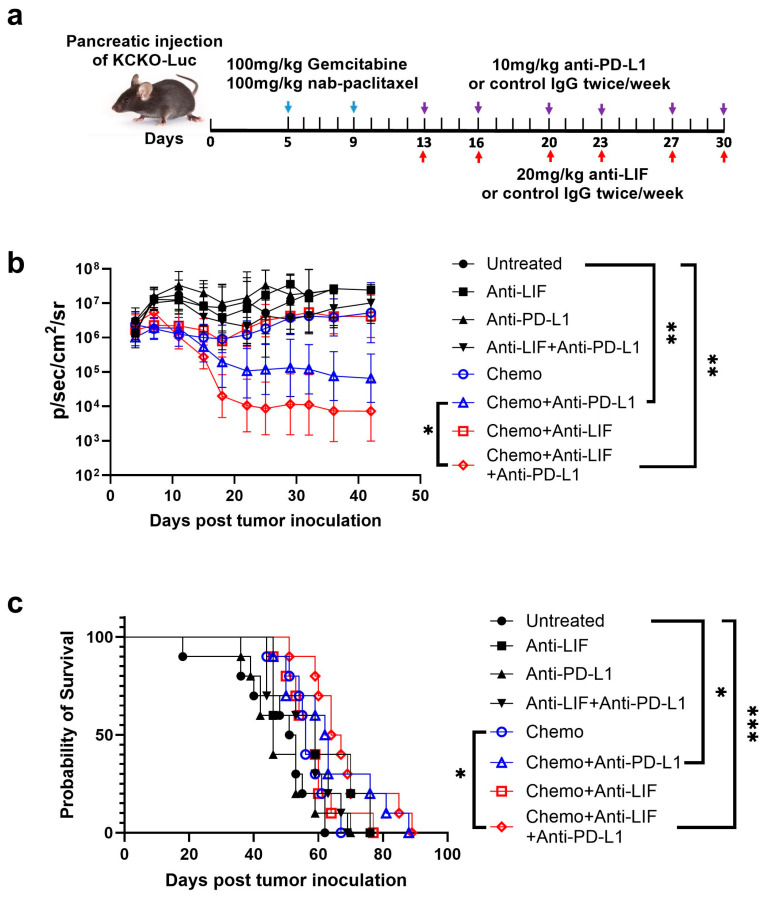
Sequential treatment of chemo and anti-LIF/anti-PD-L1 enhances antitumor efficacy. (**a**) Schematic of the experimental design. (**b**) Tumor growth curve was determined by IVIS imaging. Data shown are geometric means of the IVIS value ± SD from 10 to 15 mice per group. (**c**) Kaplan–Meier survival curve. *, *p* < 0.05, **, *p* < 0.01, ***, *p* < 0.001. IVIS, in vivo imaging system; KCKO-Luc, luciferase-expressing KCKO; PD-L1, programmed cell death-ligand 1; LIF, leukemia inhibitory factors; Chemo, gemcitabine and nab-paclitaxel.

**Figure 2 cancers-17-00204-f002:**
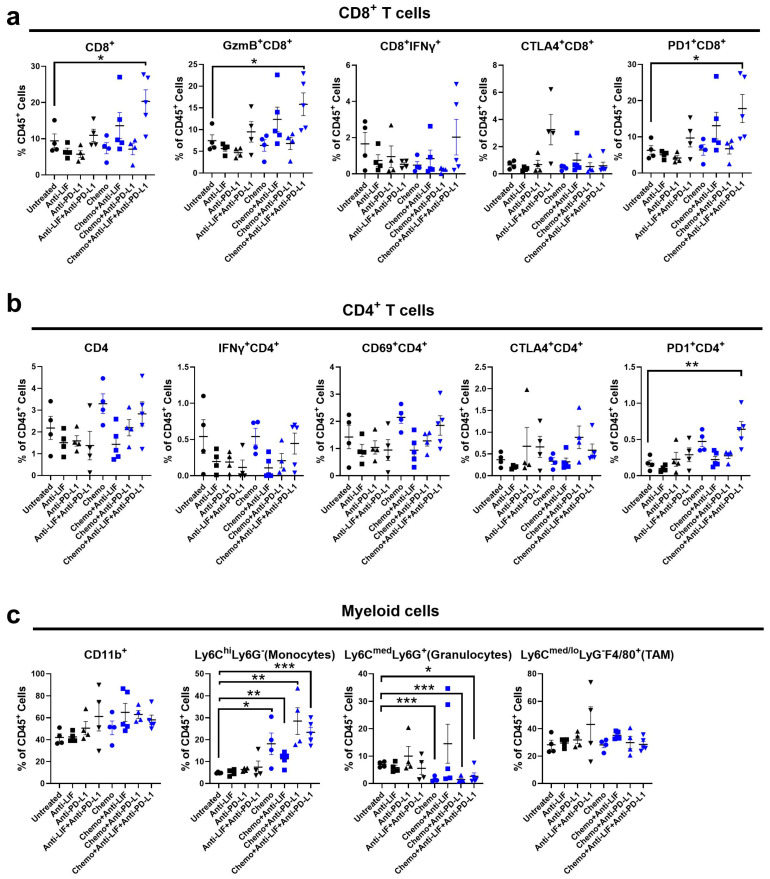
Chemo/anti-LIF/anti-PD-L1 enhances antitumor immune response in the orthotopic model of murine pancreatic cancer. Mice bearing KCKO-Luc in the pancreas were treated as in Figure 1 and sacrificed on day 15 post tumor implantation. Tumor-infiltrating immune cells were determined by flow cytometry. (**a**) Tumor-infiltrating CD8^+^ T cells and their expression of GzmB, IFNγ, CTLA4, and PD1 were analyzed. (**b**) Tumor-infiltrating CD4^+^ T cells and their expression of IFNγ, CD69, CTLA4, and PD1 were analyzed. (**c**) Tumor-infiltrating CD11b^+^ cells, monocytes (CD11b^+^Ly6C^hi^Ly6G^−^), granulocytes (CD11b^+^Ly6C^med^Ly6G^+^), and TAMs (CD11b^+^Ly6C^med/lo^Ly6G^−^F4/80^+^) were analyzed. Results are expressed as means ± SEM from 4 to 5 mice per group and analyzed by analysis of variance with Dunnett’s post-test. *, *p* < 0.05, **, *p* < 0.01, ***, *p* < 0.001. Chemo, gemcitabine, and nab-paclitaxel.

**Figure 3 cancers-17-00204-f003:**
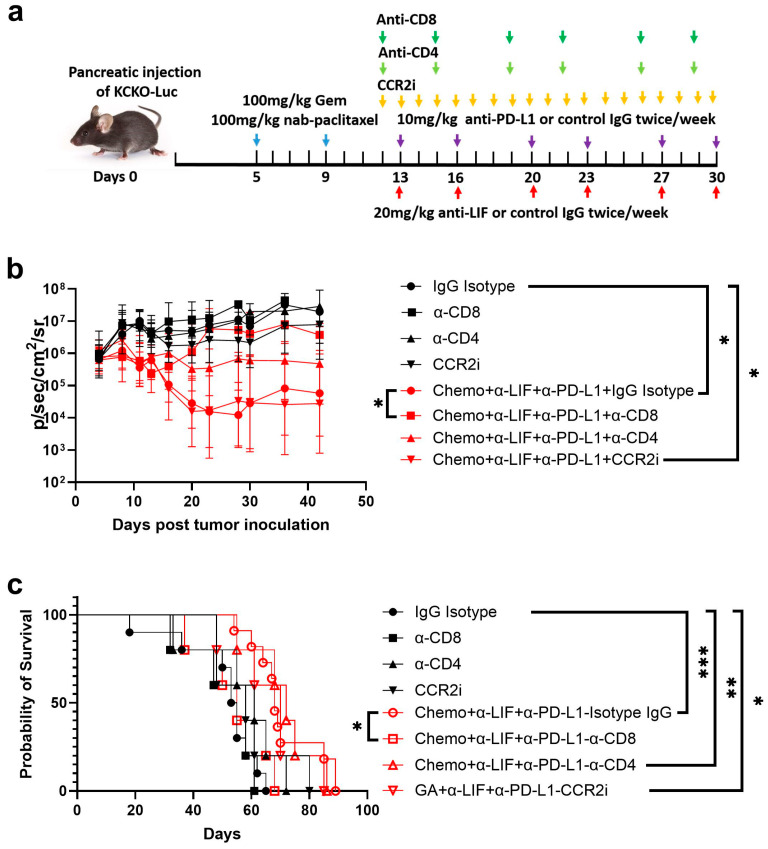
Depletion of CD8 abrogates antitumor efficacy of chemo/anti-LIF/anti-PD-L1. (**a**) Schematic of experimental design. KCKO-Luc tumor-bearing mice were treated with Chemo + anti-LIF + anti-PD-L1 as in Figure 1. Mice were given isotype or antibodies against CD8 and CD4 to deplete CD8^+^ T cells and CD4^+^ T cells, respectively. Mice were administered with CCR2i to deplete inflammatory monocytes. (**b**) Tumor growth curve. Data are represented as geometric mean ± SD (n = 5 for each group). (**c**) Survival curve. *, *p* < 0.05, **, *p* < 0.01; ***, *p* < 0.001. Chemo, gemcitabine, and nab-paclitaxel; a-LIF, anti-leukemia inhibitor factors; a-PD-L1, anti-programmed cell death ligand 1; CCR2i, C-C chemokine receptor type 2 inhibitor. Chemo, gemcitabine + nab-paclitaxel; a-CD8, anti-CD8; a-CD4, anti-CD4.

**Figure 4 cancers-17-00204-f004:**
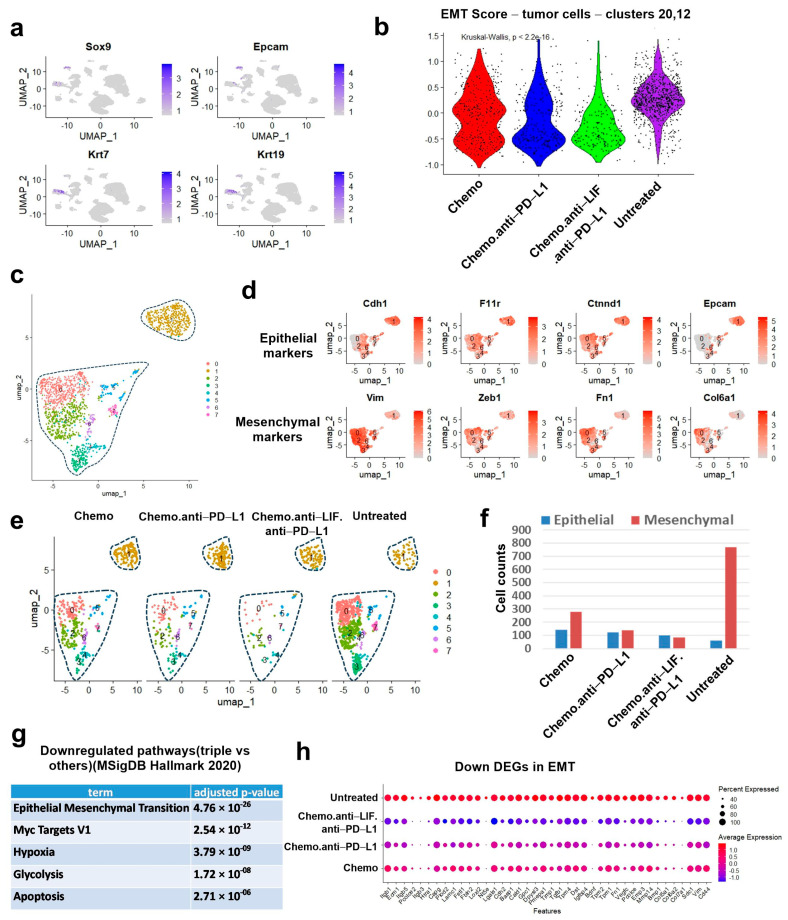
Single-cell RNA-sequencing (scRNA-seq) analysis reveals a decrease in mesenchymal tumor cells after chemo/anti-LIF/anti-PD-L1 treatment. Mice bearing KCKO-Luc in the pancreas were treated as in Figure 1a. Tumor tissues were collected on day 17 and processed for scRNA-seq. (**a**) Relative expression of tumor marker genes (Sox9, Epcam, Krt7, or Krt19) shown in UMAP feature plot. (**b**) EMT scores, which are calculated as the mean expression of the mesenchymal markers subtracted by the mean expression of the epithelial markers, shown in a violin plot. (**c**) Tumor cells (clusters 12 and 20 from the original UMAP plot, Figure A3) were reclustered. (**d**) Cluster 1 expressing high epithelial markers (Cdh1, F11r, Ctnnd1, and Epcam) was determined to be epithelial tumor cells, and other clusters expressing high mesenchymal markers (Vim, Zeb1, Fn1, and Col6a1) were determined to be mesenchymal tumor cells. (**e**) Tumor cells subclusters across different treatments, shown in UMAP plots. (**f**) Epithelial or mesenchymal tumor cell counts across different treatments. (**g**) Downregulated pathways in tumor cells (chemo.l1.lif vs. others). (**h**) Differentially expressed genes (DEGs) in tumor cells (chemo.l1.lif vs. others). Chemo, gemcitabine/nab-paclitaxel; EMT, epithelial–mesenchymal transition.

## Data Availability

All data relevant to the study are included in the article.

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
