# Peer review of "Blockade of LIF and PD-L1 Enhances Chemotherapy in Preclinical PDAC Models"

_cancers, 2025, doi:10.3390/cancers17020204_

Round 1
Reviewer 1 Report
Comments and Suggestions for Authors
Summary
In this study, Ye et al. demonstrated that the combination of chemotherapy, anti-LIF, and anti-PD-L1 can be an effective therapeutic strategy in preclinical PDAC models, and showed that it could improve anti-tumor immune response and reduced the aggressiveness of tumor cells. However, the design and experiments could be enhanced to make it more credible. My concerns are listed as below.
Major concerns:
1. In Figure 2, the authors performed flow cytometry on tumor-infiltrating immune cells at day 15 post tumor implantation. However, the schematic in Figure 1 indicates that only one dose of anti-LIF and anti-PD-L1 therapy was administered at this point, and the tumor growth curves did not differ between the combination therapy, monotherapy, and dual therapy groups at day 15. The authors should determine whether the data at this time could reflect the actual role of combination therapy on immune cells.
2. In Figure 2, the flow cytometry data was analyzed based on the proportion of total cells. It was suggested to analyze this data based on the absolute cell count or the proportion of CD45+ cells to exclude the potential influence of non-immune cells (e.g., tumor cells) on the flow cytometry results.
3. In Figure 4, the authors identified tumor cells only based on some epithelial markers. The authors should also analyze the copy number variations to more accurately define tumor cells in the scRNA-seq data.
4. Given the potential critical role of CD8+ T cells in tumor control during combination therapy, the authors should conduct a more detailed analysis of CD8+ T cells in the scRNA-seq data to elucidate their changes during therapy.
5. It is intriguing that in Figure 3 the depletion of CD8+ T cells, CD4+ T cells, and monocytes without treatment did not affect tumor control compared to the IgG isotype control. The authors should discuss the potential mechanisms involved.
Other comments:
1. More background information should be provided to explain the rationale behind including anti-PD-L1 therapy in the combination therapy.
2. The image quality needs improvement, and the font used in the figures should be consistent and adhere to the journal’s standards.
3. The authors should provide more details in the materials and methods so that others could repeat their experiments. For instance, in the Flow cytometry part, it should be specified whether the cells were stimulated prior to cytokine and granzyme detection by flow cytometry.
Author Response
Major concerns:
1. In Figure 2, the authors performed flow cytometry on tumor-infiltrating immune cells
at day 15 post tumor implantation. However, the schematic in Figure 1 indicates that only
one dose of anti-LIF and anti-PD-L1 therapy was administered at this point, and the tumor
growth curves did not differ between the combination therapy, monotherapy, and dual
therapy groups at day 15. The authors should determine whether the data at this time could
reflect the actual role of combination therapy on immune cells.
The reasons for selecting day 15 for flow cytometry analysis are as follows,
1) Timing of Immune Cell Infiltration: Significant tumor growth change in chemo+antiLIF+anti-PD0L1 group occurs between days 15 and 18. We hypothesize that changes
in immune cell infiltration, such as effector CD8+ T cells, precede the observed
changes in tumor volume and growth. Therefore, analyzing at day 15 allows us to
capture these early immune responses.
2) Tissue Availability and Feasibility: By day 18, the tumor volume in the chemo+antiLIF+anti-PD-L1 group is significantly reduced, limiting the amount of tissue available
for flow cytometry. However, regarding the reviewer’s concerns, for future
experiments, we may consider using alternative methods, such as
immunohistochemistry, to study tumor-infiltrating immune cells at later time points
when tumor volume is small.
We have added a sentence to the results section explaining our rationale for choosing the
Day 15 timepoint.
2. In Figure 2, the flow cytometry data was analyzed based on the proportion of total
cells. It was suggested to analyze this data based on the absolute cell count or the
proportion of CD45+ cells to exclude the potential influence of non-immune cells (e.g.,
tumor cells) on the flow cytometry results.
As requested by reviewer, we updated Figure 2 with proportion of CD45+ cells instead of
total cells.
3. In Figure 4, the authors identified tumor cells only based on some epithelial markers.
The authors should also analyze the copy number variations to more accurately define
tumor cells in the scRNA-seq data.
The PDAC tumor cell markers we used in Figure 4, including Sox9, Epcam, Krt7, and Krt19,
are well-established and widely used for identifying PDAC tumor cells in scRNA-seq
analyses (PMID: 3404016, PMID: 32908137, PMID: 37190324).
As suggested by the reviewer, we further analyzed copy number variations (CNVs) using
inferCNV to complement our tumor cell identification. Using cluster 1 (macrophages) as a
reference, we identified that cluster 12 (highlighted in the lower red box) exhibits significantly
higher copy numbers, particularly in chromosomes 3, 15, and 19, while cluster 20 (highlighted in the upper red box) displays lower copy numbers. These findings are consistent with the characteristics of cluster 12 as mesenchymal-like tumor cells and cluster 20 as epithelial-like tumor cells (Figure 4c-d). These data are now included in the Results Section and provided as Supplemental Figure A9.
4. Given the potential critical role of CD8+ T cells in tumor control during combination
therapy, the authors should conduct a more detailed analysis of CD8+ T cells in the scRNAseq data to elucidate their changes during therapy.
As suggested, we analyzed changes in signaling pathways (Figure A5a). Additionally, as
shown in the figure below, we assessed the relative expression of genes related to
exhaustion, proliferation, and function/effector processes in CD8+ T cells. The analysis
revealed that function/effector-related genes are relatively more highly expressed following
chemo+anti-LIF+anti-PD-L1 (chemo.lif.l1) treatment, consistent with previous flow
cytometry data. We have included these new data in Figure A5c.
5. It is intriguing that in Figure 3 the depletion of CD8+ T cells, CD4+ T cells, and
monocytes without treatment did not affect tumor control compared to the IgG isotype
control. The authors should discuss the potential mechanisms involved.
The reviewer is correct in that we did not observe significant changes in the growth curve
following the depletion of CD8+ T cells, CD4+ T cells, or monocytes in the absence of
treatment, compared to IgG isotype control groups. The inherently immunosuppressive
tumor microenvironment (TME) of PDAC inactivates intratumoral T cells, so their depletion
would have minimal effect on tumor growth in the untreated group as they are already
functionally inactive. The minimal infiltration of CD8+ and CD4+ T cells, as well as functional
T cells such as GzmB+ CD8+ and CD69+CD4+ T cells, relative to the chemo+anti-LIF+anti-PDL1 group (Figure 2a, 2b), further supports this observation. For monocytes, an
immunosuppressive population previously characterized by our group (PMID: 23653148,
PMID: 27055731, PMID: 27852031), their depletion only slightly, not significantly, reduced
tumor growth. This may be attributed to other immunosuppressive myeloid cells,
particularly tumor-associated macrophages (TAMs), which remain present and are known to
be exquisitely immunosuppressive, likely dampening the T cell response. These findings
emphasize the importance of therapies that reverse immunosuppression and/or activate T
cells against the tumor.
Other comments:
1. More background information should be provided to explain the rationale behind
including anti-PD-L1 therapy in the combination therapy.
As suggested, we emphasized in the introduction that “LIF has been identified as a novel
predictive biomarker for resistance to PD1/PD-L1 blockade in non-small cell lung cancer
(NSCLC) and bladder cancer patients, highlighting the potential of targeting LIF to enhance
ICB therapy.”
2. The image quality needs improvement, and the font used in the figures should be
consistent and adhere to the journal’s standards.
As requested, we improved the quality and adjusted the font of the figures.
3. The authors should provide more details in the materials and methods so that others
could repeat their experiments. For instance, in the Flow cytometry part, it should be
specified whether the cells were stimulated prior to cytokine and granzyme detection by
flow cytometry.
As requested, we have added additional details in the methods section including more
information regarding cell stimulation prior to flow cytometric analysis.

Reviewer 2 Report
Comments and Suggestions for Authors
In this study, the authors used Anti-LIF and Anti-PD-L1 to enhance the effect of chemotherapy in PDAC. The study is well-designed and the results are supportive. Although the study is interesting and worth publishing in Cancers, the problem is that there are no mechanisms involved in this study. All the results are descriptive. The paper might be strengthened if the authors could discuss more possible mechanisms in the discussion part.
Author Response
Comments and Suggestions for Authors
In this study, the authors used Anti-LIF and Anti-PD-L1 to enhance the effect of
chemotherapy in PDAC. The study is well-designed and the results are supportive. Although
the study is interesting and worth publishing in Cancers, the problem is that there are no
mechanisms involved in this study. All the results are descriptive. The paper might be
strengthened if the authors could discuss more possible mechanisms in the discussion
part.
We agree that summarizing the proposed mechanisms of action more clearly would
strengthen the discussion. To address this, we have revised the discussion to include new
sections elaborating on the mechanistic findings, as well as a summary paragraph at the end
that ties together the tumor-intrinsic and immune-mediated mechanisms underlying the
combination therapy. These revisions explicitly highlight how the treatment reshapes the
tumor microenvironment and enhances antitumor immune responses, addressing the
reviewer’s critique.

Reviewer 3 Report
Comments and Suggestions for Authors
The paper by Ye et al. proposed a combined therapy for pancreatic cancer. Using an orthotopic mouse model, the authors showed that treatment with chemotherapy, anti-LIF and anti-PDL1 was effective in tumor reduction as a result of CD8 T-cell activation and anti-tumor macrophages.
The manuscript is well written, and I think the work is interesting, but there are some points that need to be clarified.
MINOR POINT
-Line 249: Include the technique used to study the myeloid cell population.
-Line 288: The dosage used to deplete monocytes is already described in the Materials and methods section, it is not necessary to add it in the figure legend.
MAIN POINTS:
-Lines 211-212: Why is there no better anti-tumor effect when anti-LIF treatment is given earlier? Can the authors suggest the same mechanism or cell recruitment that is required for anti-tumor efficacy in vivo?
-Is the dose of chemotherapy used in the study similar to that used in the clinic for patients?
-Line 293-316: To confirm these data, it would be interesting to detect collagen deposition in the tumors of different treated mice by histochemical staining such as Picro Sirius or Trichomic Masson.
-The authors do not take into account the anti-tumor immunity induced by B cells. It has already been reported that patients who respond better to therapy have a humoral response and a tertiary lymphoid structure. In this combined therapy, B cells are not involved in tumor growth. I suggest to analyze the B cells infiltration in the tumor of treated and control mice.
-What about the side effects associated with the combined therapy?
-Patients who are diagnosed with PDAC usually have advanced cancer. The authors start treating mice 5 days after tumor injection, what is the stage of the tumor at that point? Is it an advanced tumor or a PanIn grade 3? What about the translatability of the proposed therapy into clinical practice?
-PDAC patients usually develop lung or liver metastases, will the proposed therapy affect the development of metastases?
-Is PD1 expressed by these tumors?
-Figure 1c: The survival of mice treated with chemo+anti-PD-L1 and chemo+anti-LIF+anti-PD-L1 seems to me very similar, which is the advantage of adding anti-LIF treatment. The same was seen in Figure 1b, the deviation bars overlap, do the authors use Anova test to analyze the differences?
Author Response
Comments and Suggestions for Authors
The paper by Ye et al. proposed a combined therapy for pancreatic cancer. Using an
orthotopic mouse model, the authors showed that treatment with chemotherapy, anti-LIF
and anti-PDL1 was effective in tumor reduction as a result of CD8 T-cell activation and antitumor macrophages.
The manuscript is well written, and I think the work is interesting, but there are some points
that need to be clarified.
MINOR POINT
-Line 249: Include the technique used to study the myeloid cell population.
As requested, we added in detailed information for myeloid cell populations in Figure 2
legend.
-Line 288: The dosage used to deplete monocytes is already described in the Materials and
methods section, it is not necessary to add it in the figure legend.
As suggested, we deleted this repeated information in the figure legend.
MAIN POINTS:
-Lines 211-212: Why is there no better anti-tumor effect when anti-LIF treatment is given
earlier? Can the authors suggest the same mechanism or cell recruitment that is required
for anti-tumor efficacy in vivo?
It remains unclear why early anti-LIF treatment was ineffective in our experiment. One
possible explanation of the observed lack of enhanced antitumor effect when anti-LIF is
administered earlier, may be due to the complex interplay between LIF inhibition and the
TME immune dynamics. Administering anti-LIF too early could disrupt the recruitment and
activation of immune cells essential for effective antitumor responses. For instance, LIF
blockade has been associated with increased infiltration of immune-supportive
macrophages and effector T-cells via mechanisms involving the CX3CR1/CX3CL1 axis.
Therefore, introducing anti-LIF prematurely might interfere with these processes, leading to
suboptimal immune cell recruitment and activation, thereby diminishing the overall
antitumor efficacy. Further research is necessary to elucidate the precise timing and
sequencing of anti-LIF administration to maximize its therapeutic potential in combination
with chemotherapy and immune checkpoint inhibitors.
We have revised the Discussion to address this point by including the following explanation:
“Interestingly, administering anti-LIF earlier, alongside chemotherapy, may disrupt the
recruitment and activation of key immune cells, such as macrophages and effector T cells,
which are essential for mounting an effective antitumor response. This highlights the
importance of optimal treatment sequencing to fully leverage the immune-stimulating
potential of anti-LIF.”
-Is the dose of chemotherapy used in the study similar to that used in the clinic for
patients?
The chemotherapy dose used in the study was determined based on a preliminary dosing
experiment, which demonstrated moderate antitumor efficacy. The gemcitabine dose was
comparable to clinical practice, while the nab-paclitaxel dose was higher than that typically
used in the clinic. This higher dose of nab-paclitaxel was necessary to achieve therapeutic
efficacy in the mouse model due to differences in drug metabolism, clearance, and tumor
biology between mice and humans.
-Line 293-316: To confirm these data, it would be interesting to detect collagen deposition
in the tumors of different treated mice by histochemical staining such as Picro Sirius or
Trichomic Masson.
Thank you for the reviewer’s valuable suggestion. We will incorporate this experiment into
our future plans.
-The authors do not take into account the anti-tumor immunity induced by B cells. It has
already been reported that patients who respond better to therapy have a humoral
response and a tertiary lymphoid structure. In this combined therapy, B cells are not
involved in tumor growth. I suggest to analyze the B cells infiltration in the tumor of treated
and control mice.
We appreciate the reviewer’s suggestion. Our analysis of B cells in the scRNA-seq data
revealed that treatments, especially chemotherapy, could decrease B cell counts, as shown
in the figures below. However, we did not observe significant changes in the signaling
pathways of B cells between the groups. Whereas the reviewer suggests that increased B
cell infiltration and tertiary lymphoid structures may support antitumor immunity, our
findings do not support this notion in the current model. B cells are not abundant in our PDAC
model, and current evidence does not strongly suggest their involvement in antitumor
immunity in this context. Further studies would be required to explore this further, but at
present, our data indicate that B cells are unlikely to play a significant role in the therapeutic
response observed.
-What about the side effects associated with the combined therapy?
As shown in the figure below, no significant body weight changes were observed in the
combined therapy group compared to the untreated group, indicating that the treatment's
toxicity is tolerable. We have included these data in Supplemental Figure A2 and referred to
the findings of this experiment in the Results Section.
-Patients who are diagnosed with PDAC usually have advanced cancer. The authors start
treating mice 5 days after tumor injection, what is the stage of the tumor at that point? Is it
an advanced tumor or a PanIn grade 3? What about the translatability of the proposed
therapy into clinical practice?
As shown in Figure 1b, by day 5 after tumor injection, the tumor is well established in the
pancreas, as detected by IVIS imaging (above 10⁶ p/sec/cm²/sr), reflecting a locally
advanced PDAC that potentially corresponds to stage II or III disease. As highlighted in the
conclusion, our findings provide strong support for ongoing and future clinical trials
evaluating the combination of ICB with anti-LIF and chemotherapy in PDAC (NCT04999969)
and propose a potential biomarker signature for this therapeutic approach.
-PDAC patients usually develop lung or liver metastases, will the proposed therapy affect
the development of metastases?
We agree with reviewer’s point and plan to evaluate the effectiveness of this combination
treatment in metastatic models in future experiments.
-Is PD1 expressed by these tumors?
We analyzed PD1 expression on T cells, as PD1 is primarily recognized as an exhaustion
marker for T cells. While PD1 expression on tumor cells has been reported in certain cancers,
it is less commonly associated with PDAC. In line with this, our flow cytometric analysis did
not detect PD1 on PDAC tumor cells in our model.
-Figure 1c: The survival of mice treated with chemo+anti-PD-L1 and chemo+anti-LIF+antiPD-L1 seems to me very similar, which is the advantage of adding anti-LIF treatment. The
same was seen in Figure 1b, the deviation bars overlap, do the authors use Anova test to
analyze the differences?
We appreciate the reviewer’s concern. For the survival analysis (Figure 1c), no significant
differences were observed between the chemo+anti-PD-L1 and chemo+anti-LIF+anti-PD-L1
groups. However, compared to the chemo group, only the chemo+anti-LIF+anti-PD-L1 group,
and not the chemo+anti-PD-L1 group, showed a significant difference. This suggests a
potential trend toward improved antitumor efficacy for the chemo+anti-LIF+anti-PD-L1
combination.
In Figure 1b, significant differences (p < 0.05) were observed between the chemo+anti-PDL1 and chemo+anti-LIF+anti-PD-L1 groups at time points 5 (day 18) and 7 (day 25). These
differences were identified using one-way ANOVA and multiple unpaired t-tests (GraphPad
Prism) to compare each time point between the two groups.

Round 2
Reviewer 1 Report
Comments and Suggestions for Authors
Accept revisions
Reviewer 3 Report
Comments and Suggestions for Authors
No further revisions are required.